# An Algorithmic Approach to Emergence

**DOI:** 10.3390/e24070985

**Published:** 2022-07-16

**Authors:** Charles Alexandre Bédard, Geoffroy Bergeron

**Affiliations:** 1Faculty of Informatics, Università della Svizzera Italiana, 6900 Lugano, Switzerland; 2Centre de Recherches Mathématiques, Université de Montréal, Montréal, QC H3T 1J4, Canada; geoffroy.bergeron@umontreal.ca

**Keywords:** emergence, Kolmogorov’s structure function, algorithmic information theory

## Abstract

We suggest a quantitative and objective notion of *emergence*. Our proposal uses algorithmic information theory as a basis for an objective framework in which a bit string encodes observational data. A plurality of drops in the Kolmogorov structure function of such a string is seen as the hallmark of emergence. Our definition offers some theoretical results, in addition to extending the notions of coarse-graining and boundary conditions. Finally, we confront our proposal with applications to dynamical systems and thermodynamics.

## 1. Introduction

Emergence is a concept often referred to in the study of complex systems. Coined in 1875 by the philosopher George H. Lewes in his book *Problems of Life and Mind* [1], the term has ever since mainly been used in qualitative discussions [2,3]. In most contexts, emergence refers to the phenomenon by which novel properties arise in a complex system which is composed of a large quantity of simpler subsystems that do not exhibit those novel properties by themselves, but rather through their collective interactions. The following citation from Wikipedia [4] reflects this popular idea: “For instance, the phenomenon of life as studied in biology is an emergent property of chemistry, and psychological phenomena emerge from the neurobiological phenomena of living things”.

For claims such as the above to have a precise meaning, an objective definition of emergence must be provided. Current definitions are framed around a qualitative evaluation of the “novelty” of properties exhibited by a system with respect to those of its constituent subsystems. This state of matters renders generic use of the term ambiguous and subjective, and hence potentially problematic within a scientific discussion. In this paper, we attempt to free the notion of emergence from subjectivity by proposing a mathematical and objective notion of emergence.

### 1.1. Existing Notions of Emergence

We review a few of the many appeals to the notion of emergence. One of them goes all the way back to Aristotle’s metaphysics [5]: “The whole is something over and above its parts, and not just the sum of them all…”. This common idea is revisited by the theoretical physicist Philip W. Anderson [6], who claims that “[…] the whole becomes not only more, but very different from the sum of its parts”. In the same essay, he highlights the asymmetry between reducing and constructing:

The ability to reduce everything to simple fundamental laws does not imply the ability to start from those laws and reconstruct the universe. In fact, the more elementary particle physicists tell us about the nature of the fundamental laws, the less relevance they seem to have for the very real problems of the rest of science, much less to those of society.

The constructionist hypothesis breaks down when confronted with the twin difficulty of scale and complexity. […] at each level of complexity, entirely new properties appear, and the understanding of the new behaviours requires research which I think is as fundamental in its nature as any other. […] At each stage, entirely new laws, concepts, and generalizations are necessary, requiring inspiration and creativity to just as great a degree as the previous one. Psychology is not applied biology, nor biology is applied chemistry.

More recently, David Wallace [7] (Chapter 2) qualifies emergent entities to be “not directly definable in the language of microphysics (try defining a haircut within the Standard Model) but that does not mean that they are somehow independent of that underlying microphysics”. Emergent objects are instead seen as patterns, and the existence of those patterns as real things is subjected to a criterion that Wallace attributes to Dennett [8]. The patterns are real if the theories that admit them in their ontology gain in usefulness, in explanatory power or in predictive reliability. For instance, Dennett’s criterion mandates that temperature be thought of an emergent but real concept because it is a *useful pattern*. In fact, temperature is not a basic entity of the microphysics, yet a full scientific description of a large system with no reference to the notion temperature completely misses a fundamental aspect. In this spirit, temperature is as real as it is useful, namely, as real as it is used in key phenomenological explanations. Useful patterns, or structures, is a central concept that is formalized within algorithmic information theory, more precisely in non-probabilistic statistics.

### 1.2. From Systems to Bit Strings

To better appreciate our reliance on algorithmic information theory, we present our epistemological standpoint. We take the realist view that there is a world outside and independent of our perception. This world is made of physical systems, and the goal of science is to understand their properties, their dynamics and their possibilities. This is done through an interplay between the formulation of theories—bold conjectures about how the world is [9]—and their experimental challenges. Pragmatically, theories can serve the purpose of providing simple models to explain the data, an idea which will be explored throughout the paper. Empirical observation, on the other hand, collects data from physical systems. But what is the nature of such data collection? How can one extract from a system, assumed to exist in reality, a string of symbols that can be taken to be binary?

The answer lies in the physics of the measurement process. Observation consists of an interaction between the physical system one cares to learn about and some prepared measurement apparatus. The measurement apparatus then interacts with a computing device (this can be the experimenter) that arranges its memory in a physical representation of a bit string *x*. A scientist collecting data about a system shall then be left with a string *x*, which, clearly, is not only determined by the investigated system. The information in *x* could reflect properties of other systems with which it has previously interacted, such as surrounding systems (the environment), the measurement apparatus and the scientist itself. As observed by Gell-Mann and Lloyd [10], this introduces several sources of arbitrariness into *x*, in addition to the level of details of the measurement and the coding convention that maps the apparatus’s configuration into bits. Furthermore, the knowledge and cognitive biases of the scientist impact *what* is being measured. For Gell-Mann and Lloyd, all this arbitrariness is to be discarded in order to define the (algorithmic) information content of the system through that of *x*. We do not share this view, as we think that this arbitrariness inhibits a well-posed definition. In fact, a subtlety of scientific investigation concerns how to probe the system in order to push into *x* the yet-to-be-understood features that it can exhibit; and this shall never be constrained by a fixed method.

Nonetheless, the subjective and error-prone connection between the physical system and the recorded data does not preclude an overall objective modelling of the world. This is because, after all, *the string originates from a real physical process*. Subjectivity and errors impact which process occurs, yet a sufficiently careful analysis of the data finds the best explanation of its origin, and so also explains the scientist’s choices and the unwanted interactions. For instance, if we ask a dishonest scientist to give us data about a system, but he elects instead to give us bits at whim, then investigating the data will lead to models of what was happening in that person’s brain, which is itself a part of reality. Thus, the string *x* is always objective data from a real system, although not necessarily the one that was presumed to be under investigation.

Once the data *x* is fixed, we face the problem of finding the best explanations for it, which is related to finding its patterns or structures. This is the main investigation of the paper. It can be done in the realm of algorithmic information theory (AIT), a branch of mathematics and logic that offers similar tools as probability theory, but with no need for unexplained randomness. Li and Vitányi, authors of the most cited textbook [11] in the field, claim that “Science may be regarded as the art of data compression” and according to the pioneer Gregory Chaitin [12], “[A] scientific theory is a computer program that enables you to compute or explain your experimental data”. Indeed, even theoretical pen and paper work constitutes symbolic manipulations which are inherently algorithmic (see Figure 1).

A point remains to be addressed. Why work with classical information and computation instead of their quantum counterparts? As quantum computation can be classically emulated [13], the quantum gain is only in speed, and not fundamental in terms of what can or cannot be computed. This work is grounded in computability theory, so by leaving aside questions of time complexity, we also leave aside quantum computation.

### 1.3. Outline

This paper is organized as follows. In Section 2, we give a review of the basic notions of algorithmic information theory, with a particular focus on non-probabilistic statistics and connections in physics. Building on those, we introduce in Section 3 an algorithmic definition of emergence and we derive from it some concepts and results. In Section 4, we illustrate the relevance of the proposed definition in a toy model as well as in applications to dynamical systems and thermodynamics.

## 2. A Primer on Algorithmic Methods

Algorithmic information theory (AIT) [14,15,16] is the mergence of Shannon’s theory of information [17] and Turing’s theory of computation [18]. Introduced in his seminal paper titled “A Mathematical Theory of Communication”, Shannon’s theory concerns the ability to communicate a message that comes from a random source of symbols. In this context, the randomness is formalized in the probabilistic setting, and represents ignorance or unpredictability of the symbols to come. The entropy is then a functional on the underlying distribution that quantifies an optimal compression of the message. Concretely, this underlying distribution is often estimated through the observed biases in the frequency of the sequences of symbols to transmit. However, noticing such biases is only a single way to compress a message. For instance, if Alice were to communicate the $10^{10}$ first digits of π to Bob, a pragmatic application of Shannon’s information theory would be of no help since the frequencies of the symbols to transmit are uniform (if π is normal, which it is conjectured to be). However, Alice could simply transmit: $$‘\mathrm{The first}\;10^{10}\;\mathrm{digits}\;\mathrm{of}\;4\sum_{n=0}^{\infty }\frac{{\left(-1\right)}^{n}}{2n+1}.’$$ Bob then understands the received message as an instruction that he runs on a universal computing device to obtain the desired message. Equipping information theory with universal computation enables message compression by all possible (computable) means—and not just via statistical biases. As we will see, the length of the best compression of a message is a natural measure of the information contained in the message.

### 2.1. Algorithmic Complexity

We give the basic definitions and properties of algorithmic complexity. See [11] (Chapters 1–3) for details, attributions and background on computability theory.

The *algorithmic complexity* K(x) of a piece of data *x* is the length of its shortest computable description. It can be understood as the minimum amount of information required to produce *x* by any computable process. Per contra to Shannon’s notion of information, which supposes an *a priori* random process from which the data has originated, algorithmic complexity is an *intrinsic* measure of information. Because all discrete data can be binary coded, we consider only finite binary strings (referred to as “strings” from now on), i.e.,
$$x\in {\left\{0,1\right\}}^{*}=\left\{\epsilon ,0,1,00,\ldots \right\}\;,$$
where ϵ stands for the empty word. For a meaningful definition, we have to select a universal computing device U on which we execute the computation to obtain *x* from the description. Such a description is called a program *p*, and since it is itself a string, the length of *p* is well defined as the number of bits in it and denoted |p|. Therefore,
$$K_{U}\left(x\right):=\min_{p}\left\{|p|:U\left(p\right)=x\right\}\;.$$ Note that abstractly, U can be thought of as a universal device in any Turing-complete model of computation. In the realm of Turing machines, a universal device expects an input *p* encoding a pair p=〈q,i〉 and simulates the machine of program *q* on input *i*. Concretely, U can be thought of as a modern computer or a human with pen and paper. This is the essence of the Church–Turing thesis, according to which all sufficiently generic approaches to symbolic manipulations are equivalent and encompass physically realizable computations. The invariance theorem for algorithmic complexity guarantees that no other formal mechanism can yield an essentially shorter description. This is because the reference universal computing device U can simulate any other computing device V with a constant overhead in program length; i.e., there exists a constant $C_{\mathrm{UV}}$ such that
$$K_{U}\left(x\right)\leq K_{V}\left(x\right)+C_{\mathrm{UV}}$$
holds uniformly for all *x*. If V is also a reference universal computing device, then $K_{U}\left(x\right)$ and $K_{V}\left(x\right)$ differ by at most a fixed constant. It is customary in this field to use the big-*O* notation and express an error term as a function of |x|=n. Henceforth, the character *n* shall be reserved to denote the length of *x*. In general, O(f(n)) denotes a quantity whose absolute value does not exceed f(n) by more than a fixed multiplicative factor, so in this case, we can write $K_{U}\left(x\right)\leq K_{V}\left(x\right)+O\left(1\right)$. Throughout the paper, O(logn) error terms shall often occur. For significant data set, |x|=n is very large and so O(logn) is comparatively a very small quantity. For instance, if α≤n, then K(α)≤O(logn) since α is a (binary) number of at most logn+1 bits, and one program to compute α is simply to enumerate all those bits.

Since the ambiguity in the choice of computing devices is lifted (up to an additive constant), we omit the subscript U in the notation. Algorithmic complexity is in this sense a *universal* measure of the complexity of *x*.

The *conditional algorithmic complexity* K(x|y) of *x* relative to *y* is defined as the length of the shortest program to compute *x*, if *y* is provided as an auxiliary input. More formally,
$$K\left(x\;|\;y\right):=\min_{p}\left\{|p|:U\left(p,y\right)=x\right\}.$$ Multiple strings $x_{1},\ldots ,x_{n}$ can be encoded into a single one denoted $\langle x_{1},\ldots ,x_{n}\rangle$. The algorithmic complexity $K(x_{1},\ldots ,x_{n})$ of multiple strings is then defined as
$$K\left(x_{1},\ldots ,x_{n}\right):=\min_{p}\{|p|:U\left(p\right)=\langle x_{1},\ldots ,x_{n}\rangle \}\;.$$

For technical reasons, we restrict the set of programs resulting in a halting computation to be such that no halting program is a prefix of another halting program, namely, the set of halting programs is a *prefix code*. One way to impose such a constraint is to have all programs *self-delimiting*, meaning that the computational device U halts its computation after reading the last bit of the program *p*, but no further. This restriction is not fundamentally needed for our purposes, but it entails an overall richer and cleaner theory of algorithmic information. For instance, the upcoming relation (Equation 1) holds within an additive constant only if self-delimitation is imposed.

A key property of entropy in Shannon’s theory is the chain rule that relates the entropy of a pair to those of the constituents. This is also achieved in the realm of AIT. Let $x^{*}$ be the shortest program that computes *x*. (If there are more than one “shortest program”, then $x^{*}$ is the fastest, and if more than one have the same running time, then $x^{*}$ is the first in lexicographic order.) Algorithmic complexity satisfies the important chain rule
(1)$$K\left(x,y\right)=K\left(x\right)+K\left(y\;|\;x^{*}\right)+O\left(1\right)\;.$$ One obvious procedure to compute the pair of strings *x* and *y* is to first compute *x* out of its shortest program $x^{*}$, and then use $x^{*}$ to compute *y*, which proves the “≤” part of (Equation 1). The “≥” side, harder to prove, states that the previous procedure to compute 〈x,y〉 is nearly optimal in terms of program length. A looser formulation of the chain rule is
(2)K(x,y)=K(x)+K(y|x)+O(logn),
where n=max(|x|,|y|).

A major drawback of algorithmic complexity for pragmatic use is that it is *uncomputable*, namely, no algorithm can return K(x) on a generic input *x*.

### 2.2. Non-Probabilistic Statistics

Standard statistics are founded upon probability theory. Remarkably, the same person who axiomatized probability theory managed to detach statistics and model selection from its probabilistic roots. Kolmogorov suggested [19] that AIT could serve as a basis for statistics and model selection for individual data. See [20] for a modern review.

In this setting, a *model* of *x* is defined to be a finite set $S\subseteq {\left\{0,1\right\}}^{*}$ such that x∈S. It is also referred to as an *algorithmic statistic* or *non-probabilistic statistic*. Any model *S* can be quantified by its cardinality, denoted |S|, and by its algorithmic complexity K(S), yielding a quantitative meaning of “simple” and “complex”. To define K(S) properly, let again U be the reference universal computing device. Let *p* be a program that computes an encoding $\langle x_{1},\ldots ,x_{N}\rangle$ of the lexicographical ordering of the elements of *S* and halts.
$$U\left(p\right)=\langle x_{1},\ldots ,x_{N}\rangle \;,\;\mathrm{where}\;S=\left\{x_{1},\ldots ,x_{N}\right\}\;.$$ Then, $S^{*}$ is the shortest such program and K(S) is its length. When *S* and $S^{\prime }$ are two models of *x* of the same complexity α, we say that *S* is *a better* model than $S^{\prime }$ if it contains fewer elements. This is because there is less ambiguity in specifying *x* within a model containing fewer elements. In this sense, more of the distinguishing properties of *x* are reflected by such a model. Indeed, among all models of complexity ≤α, the model of smallest cardinality is *optimal* for this fixed threshold of complexity.

Any string *x* of length *n* exhibits two canonical models shown in Table 1. The first is simply $S_{\mathrm{Babel}}:={\left\{0,1\right\}}^{n}$, which has small complexity as it is easy to describe—a program producing it only requires the information about *n*. However, it is a large set, containing $2^{n}$ elements. It is intuitively a bad model since it does not capture any properties of *x*, except its length. The other canonical example is $S_{x}:=\left\{x\right\}$. This time, $S_{x}$ has large complexity, namely, it is as hard to describe as *x* is, but it is a very tiny set with a single element. The model $S_{x}$ is also bad as it captures *everything* about *x*, even the noise or incidental randomness. This significantly weighs down the description of the model, and is commonly known as *over-fitting*. A good map of Montreal is not Montreal itself!

If *S* is a model of *x*, then
K(x|S)≤log|S|+O(1),
because one way to compute *x* out of *S* is to give the ⌈log|S|⌉ bit-long *index* of *x* in the lexicographical ordering of the elements of *S*, where ⌈·⌉ denotes the ceiling function. (Note that the program can be made self-delimiting at no extra cost because the length of the index can be computed from the resource *S* provided.) This trivial computation of *x* relative to *S* is known as the *data-to-model code* (which should really be called model-to-data). A string *x* is a *typical* element of its model *S* if the data-to-model code is essentially the shortest program, i.e., if
K(x|S)=log|S|+O(1). In such a case, there is no simple property that singles out *x* from the other elements of *S*. Notice also that the data *x* can always be described by a *two-part description*: the model description and the data-to-model code. Hence,
(3)K(x)≤K(S)+log|S|+O(1).

In his seminal paper [21] on the foundations of theoretical (probabilistic) statistics, Fischer stated: “The statistic chosen should summarize the whole of the relevant information supplied by the sample. This may be called the *Criterion of Sufficiency*.”. Kolmogorov suggested an algorithmic counterpart. A model S∋x is *sufficient* for *x* if the two-part description with *S* as a model is an almost-shortest description, namely,
K(S)+log|S|≤K(x)+O(logn). Here, the O(logn) refers to K(K(S),log|S|) since the self-delimited two-part code implicitly carries the length of each part as its intrinsic information, while the optimal one-part code, $x^{*}$, in general does not know about the size of each part.

Such models constrain the set of strings to those sharing “the whole of the relevant” properties that characterize *x*, which is then a typical element of those models.

Finally, a good model should not give more than the relevant information supplied by the data. The simplest sufficient model displays all the relevant properties of the data, and nothing more, thus preventing over-fitting. It is called the *minimal sufficient model* and denoted $S_{M}$.

#### Kolmogorov’s Structure Function

For a given string *x*, its associated Kolmogorov’s structure function explores trade-offs between the complexity and cardinality of possible models. This function maps any complexity threshold to the log-cardinality of the optimal model *S* within that threshold. It will be applied to investigate more interesting models, “between” $S_{\mathrm{Babel}}$ and $S_{x}$.

**Definition** **1.**
*(Structure function) The structure function of a string x, $h_{x}:N\to N$, is defined as*

$$h_{x}\left(\alpha \right)=\min_{S∋x}\left\{\left\lceil \log\right|S\right|\rceil \;:K\left(S\right)\leq \alpha \}\;.$$



We say that the optimal model of α bits or less *witness* $h_{x}\left(\alpha \right)$. Extremal points of $h_{x}\left(\alpha \right)$ are essentially determined by $S_{\mathrm{Babel}}$ and $S_{x}$, as shown in Table 1. It follows that
$$\begin{array}{c} h_{x}\left(K\left(n\right)+O\left(1\right)\right)\leq \log|S_{\mathrm{Babel}}|=n\;\mathrm{and}\;h_{x}\left(K\left(x\right)+O\left(1\right)\right)\leq \log\left|S_{x}\right|=0\;. \end{array}$$

An upper bound for $h_{x}$ is prescribed by noticing that a more complex model, $S^{\prime }$, can be built from a previously described one, *S*, by including into the description of $S^{\prime }$ the first bits of index of x∈S. In this case, for each bit of index specified, the log-cardinality of the resulting model reduces by one. This implies that the overall slope of the structure function must be ≤−1. (A knowledgeable reader may frown upon this coarse argument because prefix technicalities demand a more careful analysis as is done in Section 3.1. Such an analysis shows that the relations as presented here hold up to logarithmic fluctuations.) Applying this argument to $S_{\mathrm{Babel}}$, we conclude that the graph of $h_{x}\left(\alpha \right)$ is upper-bounded by the line n+K(n)−α.

A lower bound is obtained from applying Equation (Equation 3) to the model *S* witnessing $h_{x}\left(\alpha \right)$. In such a case, K(S)≤α and $\log|S|=h_{x}\left(\alpha \right)$, so
$$K\left(x\right)-\alpha \leq h_{x}\left(\alpha \right)\;.$$ This means that the graph of $h_{x}\left(\alpha \right)$ always sits above the line K(x)−α, known as the *sufficiency line*. The above inequality turns into an equality (up to a logarithmic term) if and only if the witness *S* is a sufficient model, by definition. Thus, this sufficiency line is reached by the structure function when enough bits of model description are available to formulate a sufficient statistics for *x*. Once the structure function reaches the sufficiency line, it stays near it, within logarithmic precision, because it is then bounded above and below by the −1 slope linear regime. The sufficiency line is always reached because $S_{x}$ qualifies as a sufficient model.

For concreteness, a plot of $h_{x}\left(\alpha \right)$, for some string *x* of length *n*, is given in Figure 2. In this example, the string *x* is such that optimal models of complexity smaller than $\alpha_{M}$ are not very insightful; they give as little information about *x* as a raw recitation of the first bits of *x*. Indeed, bits describing those models are used as inefficiently as an enumeration of *x*. In sharp contrast, $S_{M}$ is exploiting complex structures in *x* to efficiently constrain the size of the resulting set. It is fundamentally different from the optimal model of $\alpha_{M}-1$ bits as it does not recite trivial properties of *x*, but rather expresses some distinguishing property of the data. Indeed, from $\alpha_{M}$ bits of model, the uncertainty about *x* is decreased by much more than $\alpha_{M}$ bits, as *x* is now known to belong to a much smaller set. In this example, $S_{M}$ is the minimal sufficient statistics.

The complexity of the minimal sufficient statistics, $\alpha_{M}$, is known as the *sophistication* of the string *x*, which captures the amount of algorithmic information needed to grasp all structures—or regularities—of the string. Technically, here we refer to set-sophistication, as defined in [22], since sophistication has been originally defined [23] through total functions as model classes instead of finite sets. Importantly, Vitányi has investigated [24] three different classes of model: finite sets, probability distributions (or statistical ensembles, cf. the following section) and total functions. Although they may appear to be of increasing generality, he shows that they are not. Any model of a particular class defines a model in the other two classes of the same complexity (up to a logarithmic term) and log-cardinality (or analogue).

Because algorithmic complexity is uncomputable, so is the structure function. However, it can be *upper semi-computed*, which means that there is an algorithm that keeps outputting better upper bounds of the structure function until it eventually reaches the actual structure function. When this happens, the algorithm does not halt, as it keeps looking for better upper bounds, not knowing that this is in vain. In our generic context of finding explanations for observation data, this upper semi-computation can be seen to be realized by the scientific enterprise, which seeks and finds simpler and better models.

### 2.3. Algorithmic Connections in Physics

Ideas of using AIT and non-probabilistic statistics to enhance the understanding of physical concepts are not new. For example, expanding on the famous Landauer principle, Bennett [25] suggested that thermodynamics is more a theory of computation than a theory of probability, and so better rooted in AIT than in Shannon information theory. Based on his work, Zurek proposed [26] the notion of *physical entropy*, which generalizes thermodynamic entropy to ensure consistency. In the case of a system with microstate *x*, the physical entropy is defined based on a statistical ensemble *P*. The latter is very similar to an algorithmic model for *x*, except that in general a non-uniform probability distribution governs the elements of *P*, so the amount of information needed to specify an element $x^{\prime }\in P$, on average, is given by Shannon entropy $H\left(P\right)=-\sum_{x^{\prime }}P\left(x^{\prime }\right)\log P\left(x^{\prime }\right)$. Important paradoxes, such as the famous Maxwell’s demon [25,27] or Gibbs’ paradox [28], appear when it is realized that the ensemble *P*, and hence the entropy of the system, depends upon the knowledge *d* held by the agent, i.e., $P=P_{d}$. Such knowledge is usually given by macroscopic observations such as temperature, volume and pressure, and defines an ensemble $P_{d}$ by the principle of maximal ignorance [29]. However, a more knowledgeable—or better equipped—agent may gather more information $d^{\prime }$ about the microstate, which in turn defines a more precise ensemble $P^{\prime }∋x$. This leads to incompatible measures of entropy. Zurek’s physical entropy $S_{d}$ includes the algorithmic information contained in *d* as an additional cost to the overall entropy measure of the system,
$$S_{d}=K\left(d\right)+H\left(P_{d}\right)\;.$$ Note that the similarity with Equation (Equation 3) is not a mere coincidence. Zurek’s physical complexity encompasses a two-part description of the microstate. It first describes a model—or an ensemble—for it, and second, it gives the residual information to obtain from the ensemble to the microstate, on average. In fact, when the ensemble takes a uniform distribution over all its possible elements, Shannon’s entropy H(P) reduces to the log-cardinality of the ensemble, which is, up to a $k_{B}\ln2$ factor, Boltzmann’s entropy.

With sufficient data *d*, the physical entropy $S_{d}$ gets close to the complexity of the microstate K(x). The ensemble $P_{d}$ is then analogous to a sufficient statistics. Indeed, Baumeler and Wolf suggest [30] taking the minimal sufficient statistics as an objective—observer-independent—statistical ensemble (they call it *the* macrostate). Gell-Mann and Lloyd define [10] the complexity $K(P_{d})$ of such a minimal sufficient ensemble to be the *Effective Complexity* of *x*. However, because of Vitányi’s aforementioned equivalence between model classes, effective complexity is essentially the same idea as sophistication, which is why they also coincide numerically (see also [31] (lemma 21)). Finally, Müller and Szkola [31] have shown that strings of high effective complexity must have a very large logical depth, an idea to which we shall come back in Appendix A.

## 3. Defining Emergence

The previous discussion of the Kolmogorov structure function made manifest the fact that a drop like the one displayed in Figure 2 at complexity level $\alpha_{M}$ is associated with a distinguished model that accounts for meaningful properties of the data. In this example, all such properties were reflected in the description of $S_{M}$. In general, however, structure functions need not be characterized by a single drop. In this spirit, what should be thought of a string whose structure function has many drops, as displayed in Figure 3? With only a few bits of model, not much can be apprehended of *x*. With slightly more bits, there is a first model, $S_{1}$, capturing some useful properties of *x*, which leads to a more concise two-part description. Allowing even more bits, a second model $S_{2}$ is possible; while being more complex, this second model reflects more properties of *x* in such a way as to yield an even smaller two-part description. This series of models continues as the allowed complexity increases. Eventually, the structure function reaches the minimal sufficient statistics $S_{M}$, after which more complex models are of no help in capturing meaningful properties of *x*.

These observations illustrate that models can prove useful when not displaying all relevant properties of the data. Those “partial” models, while not sufficient, enable a most efficient description of the data with respect to all models of lower or equal complexity. Thus, in the same way that a model witnessing the minimal sufficient statistics is understood to capture the meaningful properties of the data, those intermediate models can be thought of as capturing only some of those meaningful properties. It is from this notion that the proposed definition of emergence is constructed.

Before going any further, a point needs to be addressed: Do strings with a structure function of many drops actually exist? Yes. In [32], it is shown that *all shapes are possible*, i.e., for any graph exhibiting the necessary properties mentioned in the previous section, there exists a string whose structure function lies within a logarithmic resolution of the graph.

The main idea of our proposal is to relate emergence to the phenomenon by which the experimental data *x* exhibits a structure function with many drops. They feature regularities that can be grasped at different levels of complexity.

### 3.1. Towards a Definition

In order to sharply define the models corresponding to drops of the structure function, and to make precise in which sense these are “new” and “understand” more properties, we construct a modified structure function upon which we formalize these notions. This shall take us to a definition of emergence.

#### 3.1.1. Index Models

As discussed briefly in the previous section, one can construct models canonically from a given model by appending to it bits of the index of the data *x* in that model.

**Definition** **2.***(Index models) For a model* S∋x *and* i∈{0,…,⌈log|S|⌉}*, the index model* S[i] *is given by the subset of S whose first i bits of index are the same as those of x.*

For concreteness, if $b_{1}b_{2}b_{3}\ldots b_{i}$ are the first *i* bits of index of *x*, one way to compute S[i] is to first execute the (self-delimiting) program that computes *S*, and then concatenate the following program: (4)$$\begin{array}{cc} \; & \underset{K(i)+O(1)\mathrm{bits}}{\underbrace{\mathrm{The}\;\mathrm{following}\;\mathrm{program}\;\mathrm{has}\;(i+c)\;\mathrm{bits}.}}\\ \; & \underset{c\mathrm{bits}}{\underbrace{\mathrm{Among}\;\mathrm{the}\;\mathrm{strings}\;\mathrm{of}\;S,\;\mathrm{keep}\;\mathrm{those}\;\mathrm{whose}\;\mathrm{index}\;\mathrm{start}\;\mathrm{with}\;}}b_{1}b_{2}b_{3}\ldots b_{i}\;. \end{array}$$ The first line of the routine is only for the sake of self-delimitation. Note that this concrete description of S[i] implies
K(S[i])≤K(S)+i+K(i)+O(1). Furthermore, for every bit of index given, the model S[i]∋x so defined contains half-fewer elements than *S* does. Hence,
$$\begin{array}{c} \log|S[i]|=\log|S|-i. \end{array}$$ As can be seen from the program displayed in Equation (Equation 4), because of self-delimitation, specifying *i* bits of index requires more than *i* extra bits of model description—it requires $\bar{\gamma }\left(i\right):=i+K\left(i\right)+c^{\prime }$, where $c^{\prime }$ accounts for the constant-sized part of program (Equation 4). An inverse of $\bar{\gamma }\left(i\right)$ can be defined as
(5)$$\bar{ı}\left(\gamma \right)=\max_{i}\left\{i:i+K\left(i\right)+c^{\prime }\leq \gamma \right\},$$
which represents the number of index bits that can be specified with γ extra bits of model description. Denoting the index model $S\left(\gamma \right):=S\left[\bar{ı}\left(\gamma \right)\right]$ and noticing that the difference between γ and $\bar{ı}\left(\gamma \right)$ is of logarithmic magnitude, one finds that
$$h_{x}\left(K\left(S\right)+\gamma \right)\leq \log|S\left(\gamma \right)|=\log|S|-\bar{ı}\left(\gamma \right)=\log\left|S\right|-\gamma +O\left(\log n\right).$$

#### 3.1.2. A Modified Structure Function

We prescribe a procedure that associates to each complexity level α a model $S^{\left(\alpha \right)}$ which has log-cardinality very close to $h_{x}\left(\alpha \right)$. Intuitively, from lower to higher complexity, α is mapped to the witness of $h_{x}\left(\alpha \right)$ whenever α follows a large enough drop of the structure function; furthermore, whenever the structure function is in a slope −1 regime, α is mapped to an index model that is derived from the last witness of $h_{x}\left(\alpha \right)$.

Formally, let $\alpha_{0}$ be the smallest complexity threshold for which $h_{x}\left(\alpha_{0}\right)$ is defined. The sequence of models $\left\{S^{\left(\alpha \right)}\right\}$ is defined recursively through
(6)$$\begin{array}{cc} \left(S^{\left(\alpha_{0}\right)},k_{\alpha_{0}}\right) & =\left(S,\alpha_{0}\right)\;\;\mathrm{with}\;S\;\mathrm{the}\;\mathrm{witness}\;\mathrm{of}\;h_{x}\left(\alpha_{0}\right)\\ \left(S^{\left(\alpha \right)},k_{\alpha }\right) & =\left\{\begin{array}{cc} \left(S,\alpha \right)\;\mathrm{with}\;S\;\mathrm{the}\;\mathrm{witness}\;\mathrm{of}\;h_{x}\left(\alpha \right)\; & \mathrm{if}\;k_{\alpha -1}+h_{x}\left(k_{\alpha -1}\right)-\alpha -h_{x}\left(\alpha \right)\geq Q\left(k_{\alpha -1}\right)\\ (S^{\left(k_{\alpha -1}\right)}\left(\alpha -k_{\alpha -1}\right),k_{\alpha -1})\; & \mathrm{otherwise}\;. \end{array}\right. \end{array}$$ To determine whether a drop at complexity level α occurs or not, and so whether a new witness of the structure function is introduced or not, the length $\alpha +h_{x}\left(\alpha \right)$ of the two-part description at complexity level α is compared to that of the two-part description permitted by the previous model, $S^{\left(k_{\alpha -1}\right)}$. If they differ by more than a threshold quantity $Q(k_{\alpha -1})$, a drop occurs. Alternatively, a drop can also be thought to occur if the structure function falls below the −1 slope regime by more than $Q(k_{\alpha -1})$. This quantity is given by
$$Q\left(\alpha \right)=K(h_{x}\left(\alpha \right)|\alpha )+O\left(\log\log n\right)\;,$$
which, for α≥K(n)+O(1), is smaller than O(logn) because $h_{x}\left(\alpha \right)\leq \left|S_{\mathrm{Babel}}\right|=n$. As a final remark about Equation (Equation 6), a unique witness of $h_{x}\left(\alpha \right)$ is determined as the model that has $\log|S|\leq h_{x}\left(\alpha \right)$ and is produced by the first program of length α to halt with output *S*.

**Definition** **3.***(Modified structure function) The modified structure function* ${\tilde{h}}_{x}\left(\alpha \right)$ *is defined as*$${\tilde{h}}_{x}\left(\alpha \right)=\log\left|S^{\left(\alpha \right)}\right|.$$

It follows from this definition that ${\tilde{h}}_{x}$ lies within an additive logarithmic term above $h_{x}$, and the two functions coincide after a drop. Why define a modified structure function ${\tilde{h}}_{x}$, which is very close to the original structure function $h_{x}$? First, a “drop” of the structure function is clearly identified: it corresponds to a point in the construction of the modified structure function where the model used is updated rather than built with bits of index. Second, Equation (Equation 6) keeps track of the actual models used at each complexity threshold as witnesses of ${\tilde{h}}_{x}\left(\alpha \right)$. They are of only two kinds, either updated to a new witness of $h_{x}\left(\alpha \right)$ or built with more bits of index. In the original structure function, for two points α and β in a slope −1 regime, nothing guarantees that the models witnessing $h_{x}\left(\alpha \right)$ and $h_{x}\left(\beta \right)$ build on the same idea. They could a priori be completely different models, capturing completely different properties about the string *x*, but it just happens that the difference of their log-cardinality is roughly β−α. However, the defining models of ${\tilde{h}}_{x}$ are constructed in a way that the −1 slope forces the models to reflect the same idea. They simply contain more or less of the index of *x*. It is the departure from the slope −1 regime in the function ${\tilde{h}}_{x}$ that indicates that a new model is used, one that intuitively captures other properties of *x*. These models shall be the ones of interest.

#### 3.1.3. Minimal Partial Models as a Signature of Emergence

We have emphasized in the construction of the modified structure function a difference between the slope −1 regime and the drops of the structure function. Indeed, while the former amounts to index models, the latter corresponds to relevant yet partial models. These will be central to our proposed definition of emergence.

The set of numbers ${\left\{k_{\alpha }\right\}}_{\alpha \in \{\alpha_{0},\ldots ,K\left(x\right)+O\left(1\right)\}}$ corresponds to the set of α’s for which there are drops in the structure function.

**Definition** **4.***(Minimal partial models) The minimal partial models are defined as the witnesses of the drops of* ${\tilde{h}}_{x}$*, namely, the models* ${\left\{S^{\left(k_{\alpha }\right)}\right\}}_{\alpha \in \{\alpha_{0},\ldots ,K\left(x\right)+O\left(1\right)\}}$ *as defined in (Equation 6).*

In what follows, we denote by $S_{1}$, $S_{2}$, …, $S_{M}$ the successive minimal partial models with respective complexity $\alpha_{1}<\alpha_{2}<\ldots <\alpha_{M}$.

**Definition** **5.**
*(Emergence) Emergence is the phenomenon characterized by observation data that display several minimal partial models.*


It can be seen that the above definition maintains the generality expected of the notion of emergence, allowing for it to be applied in many different contexts. Moreover, it will be seen to allow for a mathematical treatment of various related notions.

In view of the comments in Section 1.2, emergence is a function of the observation string *x* and not necessarily of the real object that *x* is purported to represent. For instance, in the case of the dishonest scientist who disregards the object under investigation to give bits at whim, any emergence displayed by *x* would arise from the system that produced those bits—the scientist’s brain, which is itself a part of reality.

### 3.2. Quantifying Emergence

Under the proposed definition of emergence, we develop quantitative statements. In this section, three theorems are presented. Ideas related to these results have previously been developed in algorithmic statistics [20,32,33].

To avoid many technicalities and precisions, the theorems presented in this section are formulated up to logarithmic error terms. More precise statements together with their corresponding proofs are given in Appendix A.

#### 3.2.1. The Data Specifies the Minimal Partial Models

The first theorem confirms a basic intuition. The minimal partial models should be thought of as optimized ways to give the structural information about *x*, and so it should contain almost only information about *x*. The following theorem confirms that this is the case. Most of the algorithmic information of the minimal partial models is in fact information *about x*.

**Theorem** **1.**
*Each minimal partial model $S_{i}$ can be computed from x and a logarithmic advice,*

$$K\left(S_{i}\right|x)=O\left(\log n\right)\;.$$



Using the chain rule of Equation (Equation 2) entails rephrasing the statement as $$K\left(x\right)=K\left(S_{i}\right)+K\left(x\right|S_{i})+O\left(\log n\right)\;,$$
so producing $S_{i}$ in order to get *x* is not a waste; in fact, it is almost completely a part of the algorithmic information of *x*.

**Proof.** We give a program *q* of length O(logn) that computes $S_{i}$ out of *x*.
$$\begin{array}{cc} q:\;\;\; & \mathrm{Give}\;\mathrm{explicitly}\;K\left(S_{i}\right)\;\mathrm{and}\;\left\lceil \log\right|S_{i}\left|\right\rceil \\ \; & \mathrm{Run}\;\mathrm{all}\;\mathrm{programs}\;\;p\;\mathrm{of}\;\mathrm{length}\;K(S_{i})\;\mathrm{in}\;\mathrm{parallel}\\ \; & \mathrm{If}\;p\;\mathrm{halts}\;\mathrm{with}\;U\;(p)=\langle S\rangle :\\ \; & \;\mathrm{If}\;\log|S|\leq \left\lceil \log\right|S_{i}\left|\right\rceil \;\mathrm{and}\;x\in S:\\ \; & \;\;\;\mathrm{Print}\;S\;\mathrm{and}\;\mathrm{halt}. \end{array}$$ Only the first line of *q* has a non-constant length, and since both $K(S_{i})$ and $\left\lceil \log|S_{i}|\right\rceil$ are smaller quantities than *n*, giving them explicitly requires O(logn) bits. □

#### 3.2.2. Partial Understanding

We now justify the use of the term “partial” to qualify the non-sufficient minimal partial models. Intuitively, sharp drops of the structure function should be in correspondence with non-trivial properties of the underlying string. The minimal partial models at those points should encompass an “understanding” of these properties. Naturally, the magnitude of this understanding could be equated to the size of the drop. Theorem 2 confirms this idea, when “understanding” holds the following meaning.

In the context of AIT, understanding amounts to reducing redundancy, as a good explanation is a simple rule that accounts for a substantial specification of the data. For instance, when one understands a grammar rule of some foreign language, that rule can be referred to in order to explain its many different instantiations. Those instantiations are redundant, and once the grammar rule is specified, this redundancy is reduced.

**Definition** **6.**
*(Redundancy) The redundancy of a string x of length n is defined to be*

Red(x):=n−K(x|n).



The redundancy of a string is thus the number of bits of a string that are not irreducible algorithmic information. In other words, it is the compressible part of *x*. Redundancy could then be thought of as a quantification of how much there is to be understood about *x* upon learning $x^{*}$. Comparing *x* to $x^{*}$, however, is an all-or-nothing approach, and the purpose of non-probabilistic statistics is to make sense of partial understanding by studying (two-part) programs for *x* that interpolate between the “Print *x*” and the $x^{*}$ explanations. The next definition generalizes redundancy so that it can be relative to an algorithmic model.

**Definition** **7.**
*(Randomness deficiency) The randomness deficiency of a string x, with respect to the model S is*

δ(x|S):=log|S|−K(x|S).



It measures how far *x* is from being a typical element of the set. Indeed, a typical element would have K(x|S)=log|S|+O(1) so that δ(x|S) essentially vanishes. Notice that the redundancy can be recovered from
$$\delta \left(x\right|S_{\mathrm{Babel}})=n-K\left(x\right|S_{\mathrm{Babel}})=\mathrm{Red}\left(x\right)+O\left(1\right)\;.$$ We can then explore how much each minimal partial model reduces the randomness deficiency—or understands—the data *x*. Define $d_{i}$ as the height of the drop just before getting to $S_{i}$, namely,
$$d_{i}:={\tilde{h}}_{x}\left(\alpha_{i}-1\right)-{\tilde{h}}_{x}\left(\alpha_{i}\right)\;.$$

**Theorem** **2.**
*The height of the i-th drop measures how much more $S_{i}$ reduces the randomness deficiency, compared to $S_{i-1}$, i.e.,*

$$\delta \left(x\right|S_{i-1})-\delta \left(x\right|S_{i})=d_{i}+O\left(\log n\right)\;.$$



**Proof.** Using the chain rule in Equation (Equation 2) twice, which amounts to a bayesian inversion, and Theorem 1,
(7)$$\begin{array}{ccc} \delta \left(x\right|S_{i}) & = & \log|S_{i}\left|-K\left(x\right|S_{i}\right)\\ & = & {\tilde{h}}_{x}\left(\alpha_{i}\right)-K\left(x\right)-K\left(S_{i}\right|x)+K\left(S_{i}\right)+O\left(\log n\right)\\ & = & {\tilde{h}}_{x}\left(\alpha_{i}\right)-K\left(x\right)+\alpha_{i}+O\left(\log n\right)\;. \end{array}$$With the help of Figure 4, and recalling that if γ extra bits of model description are available, $\bar{ı}\left(\gamma \right)=\gamma +O\left(\log n\right)$ bits of index can be given, observe that
$$\begin{array}{ccc} {\tilde{h}}_{x}\left(\alpha_{i-1}\right)-{\tilde{h}}_{x}\left(\alpha_{i}\right) & = & {\tilde{h}}_{x}\left(\alpha_{i}-1\right)-{\tilde{h}}_{x}\left(\alpha_{i}\right)+{\tilde{h}}_{x}\left(\alpha_{i-1}\right)-{\tilde{h}}_{x}\left(\alpha_{i}-1\right)\\ & = & d_{i}+\bar{ı}\left(\alpha_{i}-1-\alpha_{i-1}\right)\\ & = & d_{i}+\alpha_{i}-\alpha_{i-1}+O\left(\log n\right)\;. \end{array}$$ Using Equation (Equation 7),
$$\begin{array}{ccc} \delta \left(x\right|S_{i-1})-\delta \left(x\right|S_{i}) & = & {\tilde{h}}_{x}\left(\alpha_{i-1}\right)-{\tilde{h}}_{x}\left(\alpha_{i}\right)+\alpha_{i-1}-\alpha_{i}+O\left(\log n\right)\\ & = & d_{i}+O\left(\log n\right)\;. \end{array}$$ □

We can then interpret the algorithmic information in the minimal partial models as the algorithmic information of *x* that enables a reduction of the redundancy of *x*. This reduction of redundancy can be quantified by the sum of all previous drops, and the amount of redundancy left to be reduced is the sum of the drops to come. When the minimal sufficient statistic is described, with only $\alpha_{M}$ bits of the algorithmic information in *x*, the redundancy of *x* is completely reduced. The remaining information left to specify is then the index of $x\in S_{M}$, which is itself irreducible algorithmic information in *x*. However, this information is not the relevant structural information about *x*—it is the *incidental* information.

#### 3.2.3. Hierarchy of Minimal Partial Models

The following theorem shows that the algorithmic information in the minimal partial models is organized in a nested structure, namely, the information in the simpler minimal partial models is mostly contained in the more complex ones. Put differently, the complex minimal partial models can be used to compute the simpler ones with some short advice.

**Theorem** **3.**
*For j>i,*

$$K\left(S_{i}\right|S_{j})\leq \alpha_{i}-\alpha_{i-1}+O\left(\log n\right)\;.$$



The proof is relegated to Appendix A.

Note that when the structure function is very steep, the values $\left\{\alpha_{i}\right\}$ which mark the complexity of the minimal partial models in this steep regime are close to one another, so the leading term $\alpha_{i}-\alpha_{i-1}$ is small. In the limit where the (actual) structure function is so steep that it actually drops by more than $Q(\alpha_{i}-1)$ between the two neighbouring points $\alpha_{i}-1$ and $\alpha_{i}$, then the leading term vanishes, and one finds $K\left(S_{i}\right|S_{j})=O\left(\log n\right)$. This is formally stated and proved in the appendix (see Lemma A2).

### 3.3. Extending Concepts

We revisit the notions of coarse-graining and boundary conditions, broadening their scope.

#### 3.3.1. A Notion of Coarse-Graining

Many approaches to emergence appeal to some notion of coarse-graining. For instance, the relevant quantities of a physical system might correspond to functions over state space. In this case, an important tool consists in averaging those quantities over regions of that space, retaining only the large scale structures, as is done in the method of effective field theories. In the context of algorithmic statistics, coarse-graining will be seen as a special case of what we will call *regraining*. We begin by defining the notion of coarse-graining precisely.

In set theory, coarse-grainings are defined from some mother set Ω. A *partition* P of Ω is a collection of disjoint and non-empty subsets such that their union gives back Ω. Let $P_{\mathrm{fine}}$ and $P_{\mathrm{coarse}}$ be two partitions of Ω. If every element in $P_{\mathrm{fine}}$ is a subset of some element of $P_{\mathrm{coarse}}$, then $P_{\mathrm{fine}}$ is a *refinement* of $P_{\mathrm{coarse}}$ and $P_{\mathrm{coarse}}$ is a *coarse-graining* of $P_{\mathrm{fine}}$. In physics, those partitions are usually specified through non-injective functions globally defined on the state space via the pre-images.

The key point of non-probabilistic statistics is to investigate an individual object *x*, without needing to refer to other $x^{\prime }$ in the set of bit strings. Hence, algorithmic models are disconnected from the notion of partition, since a single set is defined specifically for *x*, with no requirement to define a corresponding set for $x^{\prime }$. As such, algorithmic models do not partition bit strings. Still, an algorithmic model A∋x could be qualified as a *model coarse-graining* of a B∋x if B⊆A. This type of model coarse-graining in fact occurs in the regime of index models. This motivates the extended notion of *regraining*, which is simply a change from some model A∋x to B∋x, where neither model needs to be a subset of the other. It is qualified as a *fine* regraining if |B|<|A| and a *coarse* one if |B|>|A|. Model coarse-grainings are particular cases of regrainings.

The *optimal regraining* corresponds to jumping along the minimal partial models. The optimal coarse regraining occurs in the direction from $S_{M}$ to $S_{1}$, and corresponds to modelling fewer and fewer features of the data *x* to the benefit of having simpler and simpler models. It is optimal in the sense that this procedure will yield the best possible models over all complexity thresholds. As opposed to the usual approaches, where the coarse-graining occurs by averaging over specific variables (space, momentum, time…), our notion of regraining is parametrized by theory size, and so encompasses *all computable ways* in which a coarse-graining can be done. If properties of a physical system are such that the optimal coarse regraining happens by averaging over a specific variable, like space configuration, then the algorithmic regraining shall boil down to the usual method.

#### 3.3.2. Boundary Conditions

Recall from Section 1.2 that an intrinsic difficulty of scientific investigation is that the recorded data *x* never perfectly reflects a single system. Even if we leave aside the effect of the measurement apparatus and the scientist on the data, it remains that systems are never completely isolated from an environment. As any interaction mediates an exchange of information, the effect of a large and complex environment will be modelled as random noise in models of small complexity. (An example of this situation is given by the dissipation–fluctuation theorem [34] that relates dissipative interactions in a system to the statistical fluctuations around its equilibrium point. Indeed, this theorem relates dissipation, an irreversible process that does not preserve information, with noise in the form of statistical fluctuations.) However, if the string *x* is sufficiently detailed, some structures of the environmental “noise” are grasped by models complex enough. This highlights that some information in *x* may be explained by models of great complexity but remain unexplained by a simple model. Such information can then be thought to reside outside of a simple model, namely, in its data-to-model code. This suggests that one should interpret the data-to-model code as the *boundary* of the model.

**Definition** **8.**
*(Boundary conditions) The boundary conditions of the model S corresponding to the data x are the index of x in S.*


In this definition, the scope of the term is broadened from its physical meaning, so that it can be thought of as the boundary of a model *S*, namely, the part of the system that generated the observational data *x* that is *not modelled* by *S*. The remaining structure in *x* is then viewed as coming from non-typical boundary conditions, often arising from interactions with an environment. In the case of the minimal sufficient statistics $S_{M}$, the typicality of *x* in $S_{M}$ captures the fact that the boundary conditions are arbitrary with respect to the model.

The traditional space-time boundary conditions (e.g., the initial position and velocity of a particle) of a system are an example of what is usually relegated to the data-to-model code, as models usually do not aim at explaining them. Another example is the precise values of mechanical friction coefficients. Within classical mechanics, these values come from outside the theory and would thus be a part of the boundary conditions when understood as per Definition 8. However, with more precise observations, one could explain the values of the coefficients from a more precise model that encompasses molecular interactions. More examples are provided in the following section.

## 4. Applications

The versatility of the proposed approach to emergence is now illustrated through some applications. This section is not meant to be an exhaustive review of the possible uses of these definitions, but should rather be understood as a complement to the main exposition whose purpose is twofold. We first illustrate the ideas developed with a concocted example, then we connect some of them to dynamical systems in physics.

### 4.1. Simulation of a 2D Gas Toy Model

As a first application, we consider a toy model for a non-interacting 2D gas on a lattice. The gas is taken to be spatially confined on an L×L grid with a discrete time evolution. Using a pseudorandom number generator, we choose an initial position and momentum for each of the *N* particles. Each momentum is only a direction in the set {l,r,u,d}, corresponding to left, right, up and down. The gas then evolves according to simple rules. A single particle, represented by a 1 in the lattice, just keeps its trajectory and momentum, as in Figure 5. When it bounces off a boundary, its momentum gets flipped, as in Figure 6. Intersecting trajectories are represented as displayed in Figure 7.

At any time (including the initial time), if two or more particles are at the same site, we simply write down the number of particles in the site and keep track of the momenta. As an observation *x*, we extract for each of the first *T* time steps the state in configuration space (i.e., we ignore momentum). One visual way to encode the state in configuration space is to write in each of the $L^{2}$ sites a 0, and write to the left of it, in unary, the number of particles in that site. The table of 0s and 1s, can then be expanded into a string. For instance, a 3×3 grid example of this encoding is given in Figure 8.

At each time step, the bit string corresponding to the configuration state has one 0 for each site, and one 1 for each particle, so its total length is conserved in time and thus the observational data *x* could be obtained by a mere concatenation of the *T* configuration encodings. To have an idea of the course of *x*’s structure function, models can be sought and used to determine upper bounds. In the case of this simulated 2D gas, *x* comes from the known context of the simulation, which provides important clues to find models other than the obvious $S_{\mathrm{Babel}}$ and {x}.

A first model inspired by the simulation specifies the parameters *L* and *N*, external to the gas, together with the final time *T*. Compared to {x}, which lists everything about the simulation, simplicity is gained by leaving outside of the model the initial conditions of the gas. This defines the set $S_{\mathrm{Gas}}^{L,N,T}$ of all configuration histories of *T* iterations, for each possible initial conditions of *N* particles confined to a L×L grid. The size of the shortest program for $S_{\mathrm{Gas}}^{L,N,T}$ amounts to K(L,N,T)+O(1), since the evolution rules are of constant length. Different elements of $S_{\mathrm{Gas}}^{L,N,T}$ are similar gases in different initial conditions, according to the identification of the index of $x\in S_{\mathrm{Gas}}^{L,N,T}$ with the boundary conditions.

Even simpler models can be made by pushing into the boundaries the particular values of the external parameters *L*, *N* or *T*. For illustration, we make the argument only for *T*. Let *T* be expressed by τ bits in a binary expansion. A simpler model than $S_{\mathrm{Gas}}^{L,N,T}$ can be given by producing, for each possible initial condition, *all* histories of length smaller than $2^{\tau }$. We denote this set as $S_{\mathrm{Gas}}^{L,N,<2^{\tau }}$. Its cardinality is $2^{\tau }-1$ times bigger than $S^{L,N,T}$, thus adding τ to the log-cardinality axis. If *T* was a random number,
(8)K(T)=τ+K(τ)+O(1),
and exactly τ bits would be saved on the complexity axis, since only τ, as opposed to *T*, would be needed to compute the model. In general, however, *T* is not algorithmically random—it may also contain structures. Since $S_{\mathrm{Gas}}^{L,N,<2^{\tau }}$ only encodes τ, a basic upper bound for *T*, most of the information about *T* lies outside of what is modelled; it is again the index of *x* that carries this information as boundary conditions with respect to that model. In a similar fashion, other simple models can be introduced for *L* and *N*, pushing again their information into the boundaries of the models.

The models $S_{\mathrm{Gas}}^{L,N,T}$ and $S_{\mathrm{Gas}}^{L,N,<2^{\tau }}$ provide upper bounds to the structure function. Yet, as presented in Figure 9, upper bounds to the structure function still leave one uncertain about the actual course of the structure function. In particular, in the simulation of the gas, the initial conditions were not algorithmically random, as they came from a pseudorandom number generator—a short program which, on a short inputted seed, outputs a sufficiently long string. Hence, the program in the random number generator, together with its seed, is shorter than the length of the initial conditions it generated. The actual structure function reflects this through one more drop at a higher level of complexity. The hypothetical witness of this drop, $S_{\mathrm{RNG}}$, is a model that explains the initial conditions as coming from the pseudorandom number generator. It is the set of all gas histories compatible with the dynamics previously described, and where the initial conditions have been generated with the pseudorandom program, with the seed being relegated to the data-to-model code. If the slope after $S_{\mathrm{RNG}}$ remains in a slope −1 regime, the seed is typical. However, in reality, the seed has been produced by another physical system, for instance, by the programmer. Yet again, if the seed is long enough, the structure function could reveal more drops which capture more structures, for instance, the favourite keys of the programmer who entered the seed on his keyboard. This process will go on until all that can be explained has been explained.

This example makes clear that the notion of boundary conditions really refers to a theory (or an algorithmic model), and they are fixed somewhat arbitrarily, when the users of the theory are satisfied with their notion of the system that is being modelled. In this case, if what we want to model is the gas, then $S_{\mathrm{Gas}}^{L,N,T}$ is good enough, and it is practical to declare that the initial state is typical. However, the reality may be quite different, and what we prescribe as a boundary condition to our theory may in fact be explained by a more complex, deeper theory.

### 4.2. Dynamical Systems

In this second application, we investigate how the notions introduced in this paper appear in the context of dynamical systems. We begin by documenting how the concept of integrability and chaos can be cast in the language of algorithmic information theory. This is followed by an account of how thermodynamics can be seen to emerge, under the proposed definition of emergence, from the application of statistical mechanics to complex dynamical systems.

#### 4.2.1. From Integrability to Chaos

Consider a generic classical system with Hamiltonian *H* and where the state space *M* is indexed by a set of real coordinates $X={\left\{q_{i},p_{i}\right\}}_{i\in \{1,\ldots ,\dim M/2\}}\in M$. (More precisely, *M* is a symplectic manifold parametrized locally by real coordinates forming an atlas.) Solutions to the dynamics are curves in *M* describing the evolution of the state in time. Specifying *M*, *H* and an initial point $X_{0}\in M$ singles out a unique solution curve $X_{t}$ of the dynamics. As a rudimentary formalization of some observation of the system, consider a bounded observable represented by a function *f* with f:M→[0,1]. A discrete sequence is constructed from its evaluation ${\left\{f\left(X_{j\tau }\right)\right\}}_{j\in \{1,\ldots ,N\}}$ at a regular time interval 0<τ∈Q with negligible K(τ). As this sequence is to represent a series of measurements, one must restrict its resolution, since measured values are always constrained to a finite resolution. For a real number α, we denote by ${\left[\alpha \right]}_{k}$ the truncation of its binary expansion after the first *k* bits beyond the decimal point such that ${|\left[\alpha \right]}_{k}-\alpha |\leq 2^{-k}$. This truncation effectively restricts the resolution to *k* bits as the measurement function is upper-bounded by 1. Denoting by $f_{j}^{k}\equiv {\left[f\left(X_{j\tau }\right)\right]}_{k}$ the restricted measurements, the recorded observational data string *x* is then an encoding of the sequence of measurements:$$x\equiv \langle {\left\{f_{j}^{k}\right\}}_{j\in \{1,\ldots ,N\}}\rangle .$$

We now wish to characterize the complexity of the data string *x* and study its asymptotic behaviour when the length *N* of the measurement sequence is increasing. First, one must formulate a meaningful upper bound for K(x). To that end, we require *f* to preserve information, that is,
$$K\left({\left[f\left(X\right)\right]}_{k}\;\right|\;{\left[X\right]}_{k})=O\left(1\right).$$ A trivial bound is then given by the bit length of the encoded sequence of measurements; thus,
K(x)≤kN+O(logkN). However, the regularity provided by the laws of motion implies that this bound is not strict. Indeed, given the Hamiltonian *H* and the manifold *M*, the machinery of symplectic geometry specifies the dynamical evolution as a set of differential equations that we will denote as 〈H,M〉. These equations can be integrated numerically from the initial conditions $X_{0}$ to obtain $f_{j}$ to a desired precision. These remarks, together with the stated condition on *f*, imply that
$$K\left(x\right)\leq K(\langle M,H\rangle ,\tau ,k,N,X_{0})+O\left(1\right)\;.$$ The above can be further simplified in view of studying the asymptotic behaviour in *N* by observing that the dynamical laws 〈M,H〉, the time interval τ and the resolution *k* are fixed and independent of *N*. Thus, as the length of *x* is scaled by increasing *N*, they can be taken to be constant. (To simplify the analysis, it is tacitly assumed that the dynamical laws are simple in the sense that the coefficients of the differential equations in 〈H,M〉 are of finite complexity.) Hence, one has
(9)$$\begin{array}{c} K\left(x\right)\leq K\left(N\right)+K\left(X_{0}\right)+O\left(1\right)\;. \end{array}$$ Remembering that $X_{0}$ encodes the initial conditions, which are a set of real numbers that cannot be constructively specified in general, one is left with a conundrum. Indeed, if $X_{0}$ encodes typical real numbers, the upper bound (Equation 9) is trivial as the right-hand side is infinite. However, only a finite precision in the initial conditions is required in order to integrate the system to a given precision in the final result. Thus, the resolution in $X_{0}$ required is only as much as is needed to compute ${\left\{f_{j}^{k}\right\}}_{j\in \{0,1,\ldots ,N\}}$. As such, the asymptotic behaviour of K(x) for N→∞ is determined by the scaling in the required resolution.

A chaotic dynamical system is often characterized by an exponential divergence in the evolution of nearby initial configurations, namely,
$$\frac{|X_{t}^{\prime }-X_{t}|}{|X_{0}^{\prime }-X_{0}|}=e^{\lambda t}\;,$$
where |·| denotes a metric on *M* and λ is known as the Lyapunov exponent. (It is here assumed that the Lyapunov exponent is constant and unique, which is not always the case.) In such a chaotic system,
$$|X_{0}^{\prime }-X_{0}|<2^{-\lambda^{\prime }j-k}\;\Rightarrow \;\left|X_{j\tau }^{\prime }-X_{j\tau }\right|<2^{-k}\;,\;\mathrm{with}\;\lambda^{\prime }=\frac{\lambda \tau }{\ln2}\;,$$
so *k* bits of precision on $X_{j\tau }$ can be achieved by $k+\lambda^{\prime }j$ bits of precision on $X_{0}$. Therefore, the computation of $X_{N\tau }$ from the initial condition is more efficient—in terms of description length—than straightforward enumeration if $k+\lambda^{\prime }N\leq kN$, or equivalently, if
(10)$$\lambda^{\prime }\leq k-\frac{k}{N}\;.$$ This means that for some values of the Lyapunov exponent λ, time interval τ and precision *k*, it could be more efficient to simply recite the observed data ${\left\{f_{j}^{k}\right\}}_{j\in \{1,\ldots ,N\}}$ as a genuinely random string. However, no matter how large the Lyapunov exponent is, there are time steps τ small enough to make it more efficient to calculate ${\left\{f_{j}^{k}\right\}}_{j\in \{1,\ldots ,N\}}$ from enough bits of initial conditions. Concretely, the precision on the initial conditions that can be obtained is bounded by the resolution of measurement devices. A more practical approach accounts for this with a fixed resolution $k^{\prime }>k$ in the initial conditions and is thus limited to the truncation ${\left[X_{0}\right]}_{k^{\prime }}$. This, together with the Lyapunov exponent of the system under consideration, determines an interval of predictability within which the observational data *x* can be compressed. To preserve predictability beyond this interval, one is forced to update one’s knowledge of the state of the system with a measurement. The phenomenon is well-known within chaos theory and shows up as a fundamental limitation to the predictability of such systems. Seen from the algorithmic information lens, chaos can be understood as follows: the bits of the state’s description that are initially far away from the decimal point, and hence apparently irrelevant, make their way closer to the decimal point, becoming relevant.

Dynamical systems can generally be organized by considering the asymptotic of the string of measurements *x* with N→∞. At one end of the spectrum lie integrable systems, where *k* bits of knowledge of $X_{0}$ can be used all the way through to compute the *k* bits of $f_{N}^{k}$. Those include systems where integration can be carried symbolically without an accumulation of errors. On the other side of this spectrum are chaotic systems, where $k+\lambda^{\prime }N$ bits of $X_{0}$ are required to compute the *k* bits of $f_{N}^{k}$. Similar classification schemes for dynamical systems that account for integrability and the appearance of chaos based on computational complexity have been proposed previously [35]. An algorithmic perspective on dynamical systems brings the possibility of considering other types of systems, where k+g(N) bits of $X_{0}$ can be used to compute the *k* bits of $f_{N}^{k}$, with g(N) some a priori generic function.

#### 4.2.2. Thermodynamics and Statistical Mechanics

Statistical mechanics posits the ergodicity of a complex dynamical system in order to obtain a partial, yet useful, description of its behaviour. This partial description is mostly understood to refer to the macroscopic description of a system displaying intractable microscopic descriptions. The generic approach is as follows. Starting again with a Hamiltonian and the associated phase space *M*, one first investigates the quantities conserved by the time evolution. By fixing those conserved quantities, one establishes constraints that restrict the dimensionality of the accessible phase space. Properly defined, those constraints effectively foliate the phase space into a family of submanifolds F⊆M that are each preserved by time evolution. The ergodic hypothesis now posits that the curves $X_{t}$ produced by an initial point $X_{0}\in F$ under the time evolution are dense in each submanifold *F* such that the time average value of an observable O:M→R, over such a curve is equal to the average of the same quantity over a uniform measure on each submanifold *F*,
$$\lim_{T\to \infty }\frac{1}{T}\int_{t_{0}}^{t_{0}+T}O\left(X_{t}\right)dt=\int_{F_{X_{0}}\subseteq M}O\left(X\right)d\mu \left(X\right),$$
for μ the uniform measure over *F*. This uniform measure over the submanifolds *F* is often specified indirectly in terms of the Boltzmann weights of a state X∈M over the submanifolds. With the above in mind, thermodynamics can be seen as the study of the interrelation of a relevant collection of macroscopic observables $\left\{O_{i}\right\}$, expressing the change in the value of some observables in terms of the change in the value of the others. Such a thermodynamic description of a complex system is partial yet useful and relevant to the scale at which one would like to investigate the system.

Let us now concentrate on how this very generic picture of statistical mechanics and its relation to thermodynamics fits under our proposed definition of emergence. We first define the truncation ${\left[F\right]}_{k}$ of a submanifold F⊂M to resolution *k* as the truncation of all coordinates of *F* to a *k*-bit resolution. (Technically this truncation depends on the chart, but we take an encoding of *F* to include an atlas and one for ${\left[F\right]}_{k}$ to include a prescription on the choice of the chart in which to truncate each points.) Then, positing the ergodicity of the system under study enables a direct reframing of statistical mechanics in terms of the ideas of this paper. Indeed, the postulated uniform measure on submanifolds of the phase space *M* amounts to postulating the corresponding microscopic states in a submanifold to be equally likely under time evolution. In other words, for some large enough finite time interval τ, the sequence
$$x_{N}\equiv \langle {\left\{{\left[X_{n\tau }^{\left(i\right)}\right]}_{k}\right\}}_{i\in \{1,\ldots ,\dim F\},\;n\in \{1,\ldots ,N\}}\rangle ,\;\mathrm{for}\;{X^{\left(i\right)}}_{i\in \{1,\ldots ,\dim F\}}\;\mathrm{coordinates}\;\mathrm{on}\;F,$$
is a typical sample of the truncated submanifold ${\left[F\right]}_{k}$. The lower bound on the time interval τ that needs to be satisfied for the above to hold is related to the Lyapunov exponent of the system. Indeed, such a bound corresponds to time intervals satisfying the converse of Equation (Equation 10). In such a case, $x_{N}$ is essentially an algorithmically random string. From this observation, it follows that for a sufficiently large time interval,
(11)$$K\left(x_{N}\right)=K\left({\left[F\right]}_{k}\right)+N\log\left(\right|{\left[F\right]}_{k}\left|\right),$$
which indicates that the model ${\left[F\right]}_{k}$ for the string $x_{N}$ is an algorithmic sufficient statistics.

The above discussion emphasized how thermodynamics, together with the ergodic hypothesis, amount to postulating that the models in Equation (Equation 11) associated with the decomposition into invariant submanifolds are sufficient. Indeed, a thermodynamical description of the system at equilibrium is in correspondence with such a decomposition of the phase space, provided that the conserved quantities that define the submanifolds are taken to be the thermodynamical variables. Note that ${\left[F\right]}_{k}$ is not a sufficient statistics if and only if the ergodic hypothesis fails. In such a case, one may still consider ${\left[F\right]}_{k}$ as a model of the data, but then the time sampling of the system shall not be algorithmically random; more structures can be found and incorporated to thermodynamical model.

## 5. Conclusions

We proposed a mathematical and objective definition of emergence cast in the language of algorithmic information theory. Yielding clear and far-reaching incompleteness results [36,37], this field is rich enough to mathematize mathematics itself. In this paper, we used algorithmic information theory to mathematize epistemology, giving rise to a framework which encapsulates emergence.

Intuitively, emergence is the appearance of novel properties exhibited by a complex system. In most discussions about emergence, the criteria of novelty highly depend upon the field: the aerodynamicist may be stunned by new patterns in fluid dynamics; the biochemist, by new ways in which enzymatic networks interact. In our proposed definition, emergence occurs in *“theory space”*: the thresholds of emergence are marked by the complexity of models that enable an overall shorter expression of the observed data. These models can be thought of as *understanding new structures*. Although the considered models seem very constrained (they are sets of finite bit strings), they are in fact as general as they can be since they are rooted in universal computation: any “new pattern in fluid dynamics” or “enzymatic networks interaction” that can be described is amenable to a computational process and thus an algorithmic model.

The development of our proposal was done through the minimal partial models. In Section 3, we proved that:The data specifies almost everything about the minimal partial models;The magnitude of the drop measures the amount of new understanding;More complex minimal partial models specify almost completely the simpler ones.

We also extended the notions of coarse-grainings and boundary conditions, freeing them from any specific theory. In Section 4, we considered some applications to dynamical systems and thermodynamics, and found that a new light can be shed on chaos and on the ergodic hypothesis.

The absolute generality of algorithmic-information-theoretic methods comes at the price of uncomputability. For instance, the shapes of Figure 9, in Section 4.1, are only conjectured. No program can return the structure function of a piece of data *x*. Nevertheless, the definition provides a precise framework to discuss the notion of emergence. A relaxation to the context of limited computational resources may be of interest in order to find concrete utility and applications in real-life computations; while some of the results obtained might no longer hold, the definition itself can still be applied within this limited computational context.

We recognize that the concepts involved in the proof of Theorem 3 in the upcoming Section A.3 challenge the reconciliation between our mathematical proposal and the youth of our universe. For a long string, the deep models, namely, those that occur at late drops of the structure function, are the result of programs that terminate after an unthinkably long computation. They have the largest finite running times among all programs no larger in size, so they solve the halting problem for shorter programs. This is the busy beaver regime. At a mere 14 billion years old, our universe seems too young to accommodate such computations, and this seems to hold even if we take into account the parallel computation that occurs in the observable universe. Indeed, as Bennett once put it, “the cube of Hubble’s length over Plank’s length is not even breakfast for the busy beaver!” However, assessing the actual computational capabilities of all physical phenomena in the entire universe is an open problem. Another way out of this conundrum is to leave the deep information in the initial conditions of the universe. One could then ask about the source of this information. Yet another possibility is that systems in nature are confined to relatively shallow models.

Facing the realization that models witnessing drops of the structure functions are made of halting information, Vereshchagin and Shen [20] wrote “This looks like a failure. […] [I]f we start with two old recordings, we may get the same information [about their minimal sufficient statistic], which is not what we expect from a restoration procedure. Of course, there is still a chance that some Ω-number [halting information] was recorded and therefore the restoration process indeed should provide the information about it, but this looks like a very special case that hardly should happen for any practical situation.” Facing this, they suggest considering models of more restricted classes or adding some additional conditions theirby looking for “strong models”. On the contrary, we think that that the minimal sufficient statistics of two recordings *should* share information, as they inevitably share a very common origin, which the model aims to capture. That this shared information is about the halting problem simply reflects the fact that their plausible common origin is the fruit of a very long computation, and not that the recording has anything to do with an Ω-number, or any direct representation of the halting problem.

Let us conclude with philosophy of science. Consider the string *x* to be an encoding of all scientific data ever recorded. Scientific theories aim at explaining *x* by grasping patterns in the data in order to reduce its redundancy. They are expressed in terms of models that distillate the structures from the apparently noisy boundary conditions. In an effort to tell apart theories or through pure experimental curiosity, the data *x* always increases. Through conjectures, scientists suggest new models that can better explain the observations. Better theories arise; they find important structures in apparent noise, or unify different models under the same umbrella. These models can either be proven false by some eventually contradicting piece of data, or discarded when a simpler and fitter model is found, namely, one that sits closer to the structure function. *Hence the process of scientific investigation may be identified with the upper semi-computation of the structure function of our observations*. The computational impossibility of deriving the optimal models from the data is reminiscent of the non-existent scientific induction. Theories are conjectured as tentative best explanations: they can never be proven right, nor optimal, nor final, as mandated by *fallibilism*.

## Figures and Tables

**Figure 1 entropy-24-00985-f001:**
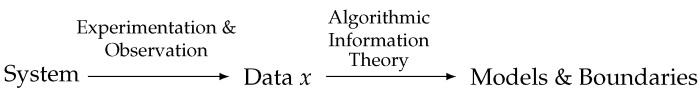
Systems are comprehended through experimentation and observation, which physically yield a bit string. Models and their respective boundaries can then be defined for each string through methods from algorithmic information theory.

**Figure 2 entropy-24-00985-f002:**
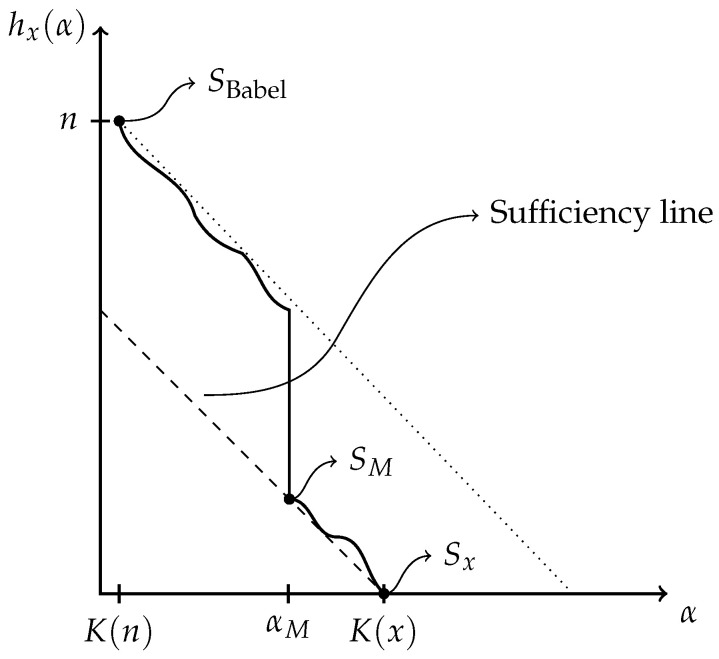
Kolmogorov’s structure function of a string *x* of length |x|=n.

**Figure 3 entropy-24-00985-f003:**
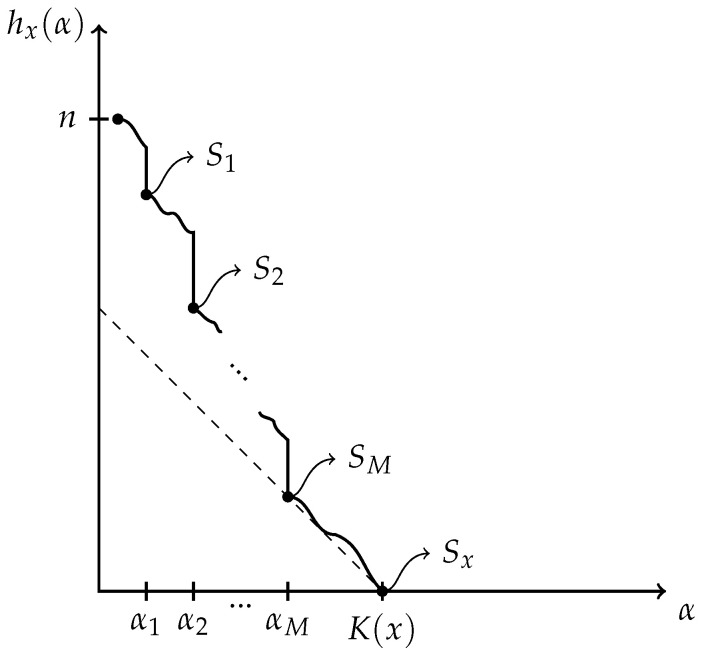
A structure function with many drops.

**Figure 4 entropy-24-00985-f004:**
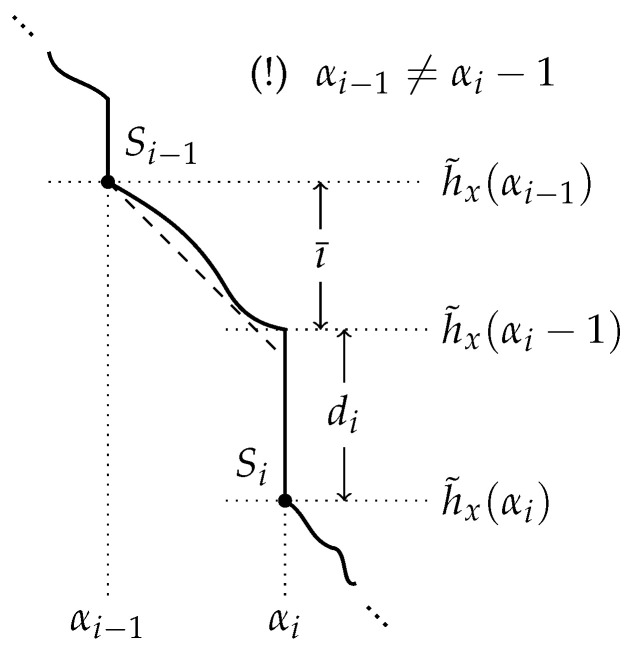
A visual help for the proof of Theorem 2.

**Figure 5 entropy-24-00985-f005:**
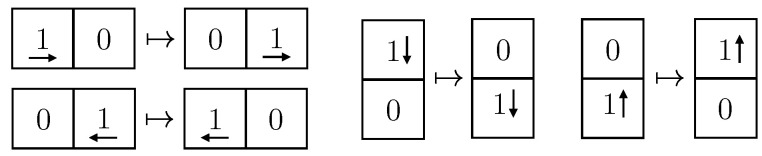
A gas particle freely moving.

**Figure 6 entropy-24-00985-f006:**
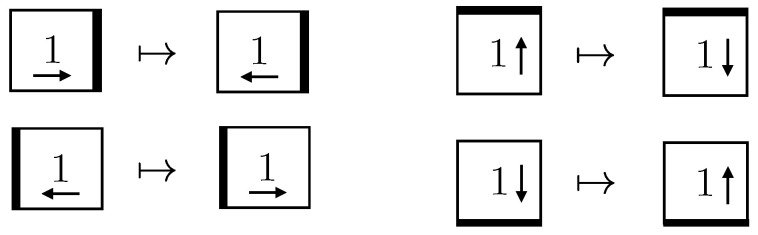
A particle bouncing off walls.

**Figure 7 entropy-24-00985-f007:**
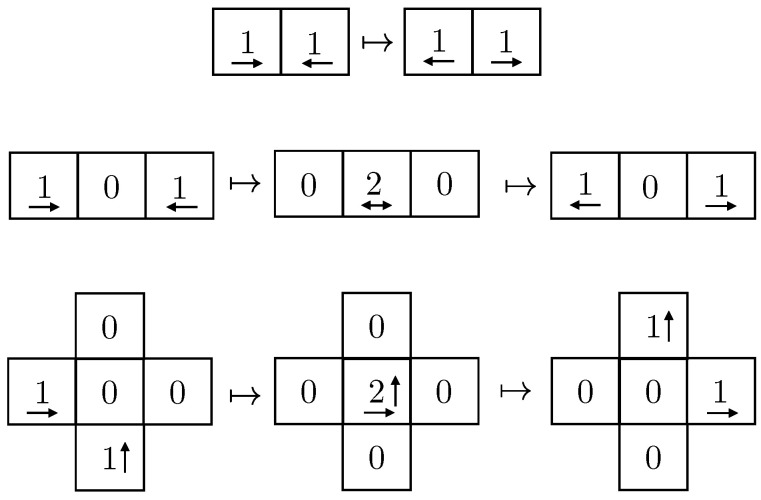
Particles “collide” as if they go through one another.

**Figure 8 entropy-24-00985-f008:**
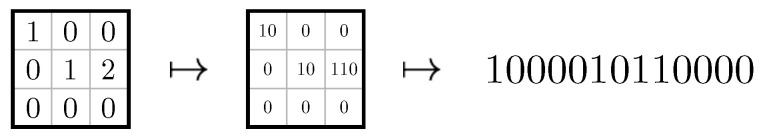
Encoding of the configuration state into bits.

**Figure 9 entropy-24-00985-f009:**
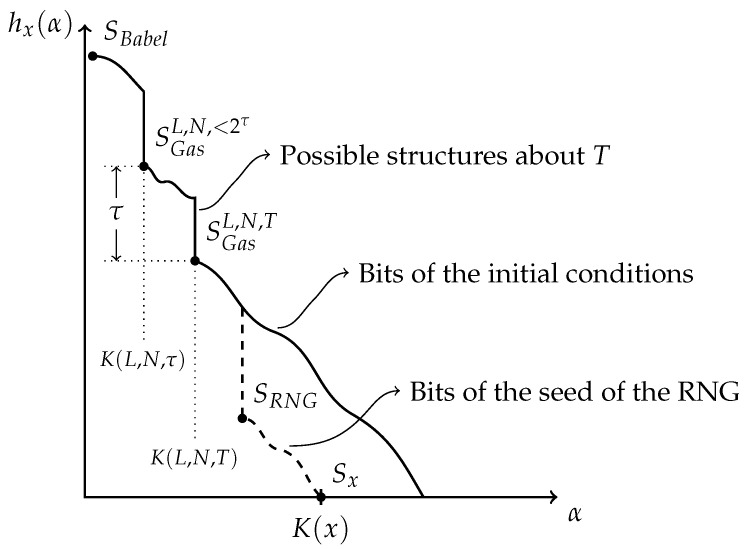
The solid line represents the known upper bound of the structure function. The dashed line represents the hypothesized real structure function.

**Figure A1 entropy-24-00985-f0A1:**
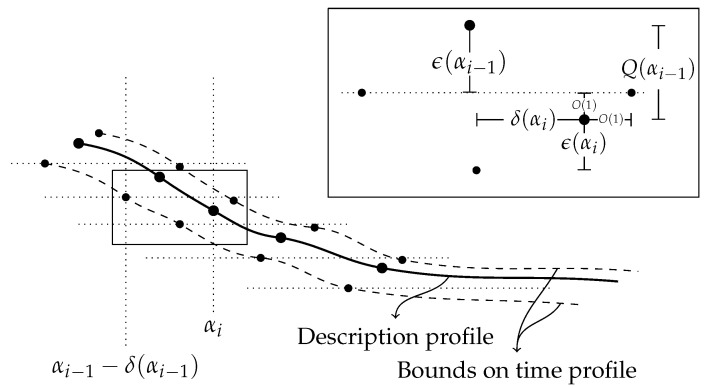
A visual help for the proof of Theorem 3.

**Figure A2 entropy-24-00985-f0A2:**
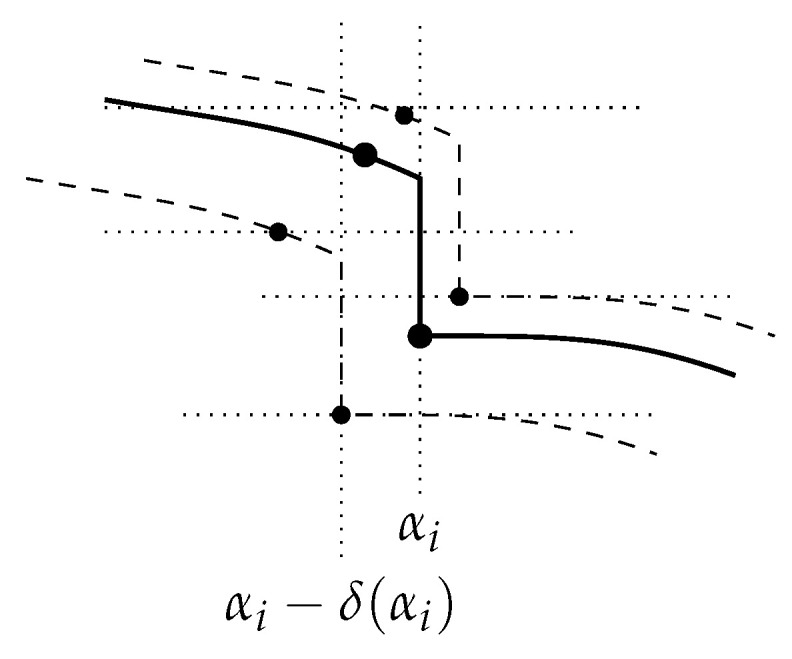
A visual help for the proof of Lemma A2.

**Table 1 entropy-24-00985-t001:** Complexity and cardinality of $S_{\mathrm{Babel}}$ and $S_{x}$.

Model *S*	Complexity	K(S)	Cardinality	log|S|
$S_{\mathrm{Babel}}={\left\{0,1\right\}}^{n}$	Small	K(n)+O(1)	Large	*n*
$S_{x}=\left\{x\right\}$	Large	K(x)+O(1)	Small	0

## Data Availability

Not applicable.

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
