# Peer review of "An Algorithmic Approach to Emergence"

_entropy, 2022, doi:10.3390/e24070985_

Round 1

Reviewer 1 Report

The reviewed paper proposes a formal explication of the notion of emergence discussed in philosophy of science or in complex systems. Quite naturally, this proposal is rooted in algorithmic statistics and, in particular, it applies the concept of Kolmogorov's structure function. The Kolmogorov structure function is a function which given the minimal description length of a model for some data returns the minimal lower bound for the description length of the data given the model. To make this bound simple, it is assumed that admissible models for data are finite supersets containing the data, whereas the conditional description length of the data is approximated by the log-cardinality of the model.

As the authors argue, in such a framework, the notion of emergence can be understood as the existence of a sequence of nested partly optimal models for the data, corresponding to different levels of understanding patterns in the data. In turn, these partly optimal models correspond to sudden drops of Kolmogorov's structure function. This is the basic and quite natural thesis of the authors. The paper explores its consequences in detail, although the investigations are mostly intellectual, since Kolmogorov complexity is well-known to be approximable from above ultimately but in an uncontrolled way.

The manuscript is clearly written but intellectually demanding. It presents a general theory and a few new theorems concerning the levels of data description. Moreover, it illustrates their applications in several thought experiments. Especially intriguing are links between (two-part) description profiles and time profiles of individual strings.

The reviewed paper contributes to a field that could be called algorithmic philosophy of science. But what it proposes is just the beginning, since it revolves around examples only from natural sciences. Indeed, natural sciences are a classical playground from philosophy of science to test the proposed framework. However, to make the picture more complete, let me try to argue in the following that language sciences may be also an important source of inspiration for theoretical phenomena to be investigated in this domain.

In fact, texts in natural language constitute an interesting and important playground on which algorithmic statistics could be tested, as a thought experiment of course. They are sequences of discrete symbols and are largely available to processing by computers---an ideal environment for wild speculations about their Kolmogorov complexity!  Forming a theory of texts in natural language is a joint enterprise of linguistics and artificial intelligence. However, theorizing about language is complicated. Linguists differentiate various levels of language description such as phonology, morphology, syntax, semantics, and pragmatics. Theories of particular of these levels interact among each other rather strongly. Would these interactions correspond to blurred drops of the structure function for language data?

An important issue in this context, not discussed by the authors, is:

What happens to the Kolmogorov complexity and structure function if we keep on enlarging the size of the data? Does the minimal sufficient model grow with the amount of data?

The authors might have heard of large scale neural statistical language models such as GPT-{2,3} or BERT. These models are trained to minimize the cross entropy measure, which is an upper bound for the Kolmogorov complexity of the text. At the present state of the art, they are capable of generating intelligible random short stories. Experiments with these neural statistical language models suggest that the size of the optimal model grows with the amount of text at a power-law rate.  This phenomenon can be hypothetically linked with Zipf's law on the level of algorithmic information theory by the concept of a perigraphic process. In short, perigraphic processes are stationary stochastic processes that describe a fixed algorithmically random sequence at a power-law rate. This phenomenon implies a power-law decay of conditional entropy of the process. I wonder whether it also implies a power-law growth of the sophistication of strings generated by such a source (in expectation).

An important remark made by the authors is:

"Before going any further, a point needs to be addressed: Do strings with a structure function of many drops actually exist? Yes. In [31], it is shown that all shapes are possible, i.e., for any graph exhibiting the necessary properties mentioned in the previous section, there exists a string whose structure function lies within a logarithmic resolution of the graph."

In particular, the following questions arise: Does Zipf's law blur the drops of the structure function for language data implied by distinct levels of language description? Do we have many little drops instead of a few big ones corresponding to many little realizations of understanding rather than one big theory of everything? Can the drops of structure function merge or split as we increase the data size? What phenomenon does it correspond to at the level of philosophy of science? What mergers or splits are allowed?  Can we model some effects of deception when wrong/toxic data are concatenated with correct data?

The above comments might be used by the authors, as they wish, by reacting to them in this or in a future work. I hope that they may be inspiring. In any case, I consider the present manuscript essentially complete to be published.

Minor issues:

line 108: connexions -> connections (?)

line 627: open -> conclude (?)

Author Response

Thank you very much, dear reviewer.

You wrote:

'What happens to the Kolmogorov complexity and structure function if we keep on enlarging the size of the data? Does the minimal sufficient model grow with the amount of data?'

Good questions. More explorations can be made on the topic, but as initial thoughts, I think that one can make both cases possible, namely, an increasing complexity of the minimal sufficient model and a static one. The first case could occur if the data of an emergent, yet somewhat simple, system is first observed, and then data from a much larger and more complex system are appended. The second case can occur if more data of the same emergent system are given after the minimal sufficient statistics has been grasped. In such a case, all the new data falls into incidental randomness.

Thanks for the other pointers. We'll check them up.

Reviewer 2 Report

A review of

An Algorithmic Approach to Emergence

written by

Charles Alexandre Bédard, Geoffroy Bergeron

This is a very interesting paper that uses Kolmogorov complexity to describe and quantify the notion of emergence. The phenomenon of emergence is what happens when an ensemble combines. The ensemble has properties that were not present in the original system. The authors use Kolmogorov complexity theory and Kolmogorov’s structure map to describe such emergent properties. The idea is not easy to describe but seems correct.

The paper is not only theoretical. Section 4 of the paper shows how this work can be used to understand toy models of 2D gas, dynamical systems, and certain aspects of statistical mechanics.

Section 5 has some philosophical ideas about the work. It is important to note that since Kolmogorov complexity is uncomputable, some of this work is uncomputable. This does not detract from the work. Just because the Halting problem is uncomputable does not mean we should abandon computability theory.

Although the paper is fairly complex, the authors have put some very technical parts in an appendix at the end of the paper. This was wise and adds to the readability of the main part of the text.

The paper is well-structured and tries to motivate its work. The authors are clear and exact. I found no typos and the paper can be published in its present form.

In conclusion, while I think the paper is very complicated, I strongly urge the journal to publish this important work.   

Author Response

Thanks :)